# Temperature dependence of quantum oscillations from non-parabolic dispersions

Chunyu Guo [1,11✉], A. Alexandradinata[2,3,4,11✉], Carsten Putzke [1], Amelia Estry[1], Teng Tu[5], Nitesh Kumar [6], Feng-Ren Fan[6], Shengnan Zhang[7,8], Quansheng Wu [7,8], Oleg V. Yazyev [7,8], Kent R. Shirer[6], Maja D. Bachmann[6,9], Hailin Peng [5], Eric D. Bauer [10], Filip Ronning[10], Yan Sun [6], Chandra Shekhar [6], Claudia Felser[6] & Philip J. W. Moll [1✉]

The phase offset of quantum oscillations is commonly used to experimentally diagnose topologically nontrivial Fermi surfaces. This methodology, however, is inconclusive for spin-orbit-coupled metals where $\pi$-phase-shifts can also arise from non-topological origins. Here, we show that the linear dispersion in topological metals leads to a $T^2$-temperature correction to the oscillation frequency that is absent for parabolic dispersions. We confirm this effect experimentally in the Dirac semi-metal $Cd_3As_2$ and the multiband Dirac metal $LaRhIn_5$. Both materials match a tuning-parameter-free theoretical prediction, emphasizing their unified origin. For topologically trivial $Bi_2O_2Se$, no frequency shift associated to linear bands is observed as expected. However, the $\pi$-phase shift in $Bi_2O_2Se$ would lead to a false positive in a Landau-fan plot analysis. Our frequency-focused methodology does not require any input from ab-initio calculations, and hence is promising for identifying correlated topological materials.

[1] Laboratory of Quantum Materials (QMAT), Institute of Materials (IMX), École Polytechnique Fédérale de Lausanne (EPFL), CH-1015 Lausanne, Switzerland. [2] Institute for Condensed Matter Theory, University of Illinois at Urbana-Champaign, Urbana, IL 61801, USA. [3] Department of Physics, University of Illinois at Urbana-Champaign, Urbana, IL 61801, USA. [4] Physics Department, University of California Santa Cruz, Santa Cruz, CA 95064, USA. [5] Center for Nanochemistry, Beijing National Laboratory for Molecular Sciences (BNLMS), College of Chemistry and Molecular Engineering, Peking University, 100871 Beijing, China. [6] Max Planck Institute for Chemical Physics of Solids, 01187 Dresden, Germany. [7] Chair of Computational Condensed Matter Physics (C3MP), Institute of Physics (IPHYS), École Polytechnique Fédérale de Lausanne (EPFL), CH-1015 Lausanne, Switzerland. [8] National Centre for Computational Design and Discovery of Novel Materials MARVEL, École Polytechnique Fédérale de Lausanne (EPFL), CH-1015 Lausanne, Switzerland. [9] School of Physics and Astronomy, University of St Andrews, St Andrews KY16 9SS, UK. [10] Los Alamos National Laboratory, Los Alamos, NM 87545, USA. [11] These authors contributed equally: Chunyu Guo, A. Alexandradinata. ✉email: chunyu.guo@epfl.ch; aalexan7@illinois.edu; philip.moll@epfl.ch

The discovery of topological semimetals promises an avenue to study novel materials that host quasiparticles that mimick relativistic Dirac and Weyl fermions in high-energy physics[1–4]. They host bands that touch at points or lines in momentum space; such degeneracies are typically associated with closed Fermi surfaces with topologically robust Berry phases. While initially considered a rare occurrence, recent ab-initio programs[4–7] have predicted topological band degeneracies in a sixth of all non-magnetic materials in the crystal database[5].

Magnetic quantum oscillations[8] promise to play a key role in experimentally confirming these predictions. It is widely believed that a $\pi$-phase shift in quantum oscillations is interpretable as a $\pi$ Berry phase, and therefore a smoking-gun confirmation of a topological semimetal. Such phase analysis, often carried out with a 'Landau-fan plot', ignores other non-geometric phase shifts, and forgets that the Berry phase is not quantized to an integer multiple of $\pi$ for many symmetry classes of (semi)metals[9,10]. For this reason, an unambiguous topological diagnosis is generally impossible for the 3D Dirac semimetals[1–3] and low-symmetry 3D Weyl semimetals[9].

For these cases, we present a new identification method based on the temperature ($T$) dependence of the oscillation frequency ($F$), finding a characteristic $T^2$ contribution that is uniquely attributed to the linear energy-momentum dispersion of Dirac, Weyl and multifold fermions[11]. This $T^2$ contribution is a 3D, higher-degeneracy generalization of an effect predicted by Kübbersbusch and Fritz[12] for 2D Dirac materials. To the best of our knowledge, this effect has never been experimentally studied in graphene, yet we will show that it is easy to see in 3D Weyl/Dirac materials. Our strategy applies to candidate topological Fermi pockets which are small compared to the Brillouin-zone volume; small pockets are accurately described by $\mathbf{k} \cdot \mathbf{p}$ Hamiltonians that retain only the leading-order term—giving a parabolic dispersion for the Schrödinger-type fermion, and a linear dispersion for the Dirac-type fermion. These two cases are distinguishable by the energy derivative of the cyclotron mass $m_c$ (Fig. 1). While $\partial m_c/\partial E = 0$ for a parabolic dispersion, for a linear dispersion $E(k) = \pm \sqrt{(v_x \hbar k_x)^2 + (v_y \hbar k_y)^2}$, the particular energy dependence of the Fermi-surface area $S$ yields a non-zero energy derivative of the cyclotron mass:

$$S = \frac{\pi}{\hbar^2} \frac{E_F^2}{v_x v_y} \;\Rightarrow\; m_c = \frac{\hbar^2}{2\pi} \left| \frac{\partial S}{\partial E} \right| = \frac{|E_F|}{v_x v_y} \;\Rightarrow\; \frac{1}{m_c} \left| \frac{\partial m_c}{\partial E} \right| = \frac{1}{|E_F|}, \tag{1}$$

with $E_F$ the Fermi energy measured from the Dirac point and $v_j$ the Fermi velocity.

To experimentally determine $\partial m_c/\partial E$, we exploit the fact that quantum oscillations probe the band structure over an energy window ($\sim k_B T$) around the Fermi level, due to the thermal broadening of the Fermi-Dirac distribution function. As lighter-mass particles experience less thermal damping than heavier particles, the effective frequency renormalizes towards lighter orbits as $T$ increases. This effect is absent for parabolic bands because the effective mass is energy-independent. However, for a Dirac-type Fermi surface, the frequency decreases with increasing $T$ because the effective mass is smaller closer to the node, as illustrated in Fig. 1a. This temperature-renormalization of $F$ applies generally to linear dispersions near band degeneracies, including Dirac and Weyl degeneracies, as well as higher-fold degeneracies associated with Fermi pockets with higher Chern numbers (so-called 'multifold fermions')[11].

## Results

**General methodology.** To quantify this frequency shift, it is useful to view the Lifshitz–Kosevich formula[13,14] as an asymptotic expansion in powers of $k_B T/E_F$ (the degeneracy parameter of Fermi gases). Odd powers of $T$ modify the oscillation amplitude, with the first odd power giving the well-known thermal damping factor; in contrast, even powers of $T$ modify the frequency and phase. At elevated temperatures where $2\pi^2 k_B T$ is large or comparable to the cyclotron energy, we derive in supplementary note 2(A) a $T^2$-correction to the oscillation frequency:

$$F_0(\mu) \to F(\mu, T) = F_0(\mu) + \Delta F^{\text{top}}(T),$$
$$\Delta F^{\text{top}}(T) := -\frac{\pi^2}{4} \frac{(k_B T)^2}{\beta} \left| \frac{\partial (\log m_c)}{\partial E} \right|, \tag{2}$$

with $\mu$ the chemical potential and $\beta := e\hbar/2m_c$ the effective Bohr magneton. The correction $\Delta F^{\text{top}}$ is our main theoretical result, and applies to any closed Fermi surface originating from an arbitrary energy-momentum dispersion, whether linear, quadratic, cubic or beyond. While $\Delta F^{\text{top}}$ vanishes for parabolic bands, it is finite for linear bands as $|\partial(\log m_c)/\partial E| = 1/|E_F|$, according to Eq. (1), and we hence refer to $\Delta F^{\text{top}}$ as a topological correction to $F$.

Next we consider other mechanisms for the temperature dependence of the frequency. A distinct $T^2$ correction arises from the temperature dependence of the chemical potential [$\mu(T)$] at fixed particle density, as is well known in the Sommerfeld theory of metals[15] (Fig. 1b). The sum of Sommerfeld and topological

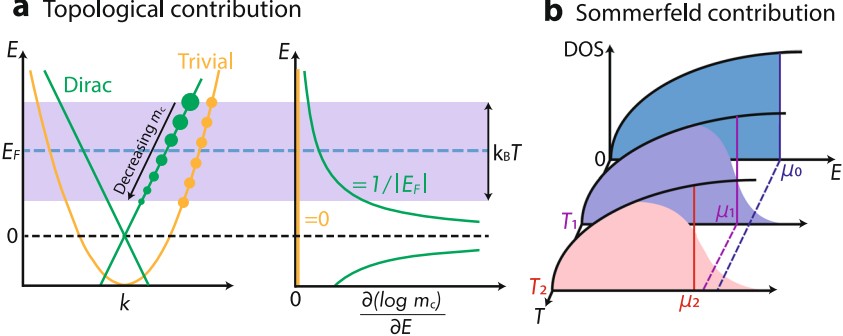

**a** Topological contribution **b** Sommerfeld contribution

**Fig. 1 Illustration for topological and Sommerfeld contributions to temperature dependence of oscillation frequency. a** For a linearly dispersing Dirac-type pocket, the energy derivative of the cyclotron mass, $\partial(\log m_c)/\partial E$ diverges when the Fermi level approaches the Dirac node. When approaching the Dirac node, the Fermi pocket shrinks and the cyclotron mass is continuously decreasing to zero, therefore the smaller the oscillation frequency, the larger the oscillation amplitude. Due to the thermal broadening of chemical potential, this ultimately leads to the quadratic temperature dependence of the quantum-oscillation frequency. In contrast, for a Schrödinger-type pocket with a parabolic dispersion, $\partial(\log m_c)/\partial E = 0$. **b** Illustration of Sommerfeld contribution, describes the shift of chemical potential at finite temperatures due to thermal broadening with a fixed carrier density.

corrections is expressed in:

$$F(\mu, T) = F_0(E_F) - \Theta \frac{(\pi k_B T)^2}{\beta^2 F_0(E_F)} + O(T^4), \qquad (3)$$

with $\Theta$ a dimensionless coefficient. In high-carrier-density metals with both small and large pockets, the Sommerfeld correction is negligible. The frequency shift of a small pocket is reduced by a factor $|E_F|/E_{bw} \ll 1$, with $E_F$ the Fermi energy of the small pocket measured from band extremum/degeneracy, and $E_{bw}$ the typical bandwith. Thus we expect that $\Theta$ is dominated by the topological correction, giving $\Theta \approx 0$ in the parabolic case, and $\Theta \approx 1/16$ in the case of a linear dispersion. For single-frequency, low-carrier-density semimetals, the Sommerfeld correction is of the same order of magnitude as the topological correction, giving $\Theta = 1/48$ for parabolic bands and $\Theta = 5/48$ for linear bands [see supplementary note 2(B) for extended calculation of Sommerfeld correction]. The two scenarios show that the observation of a $T^2$ correction alone is not conclusive of nontrivial topology. Rather, conclusiveness comes from the following experimental consistency check: since $F_0(E_F)$, $m_c$ and additionally the frequency shift at elevated temperatures all can be experimentally determined, by applying Eq. (3) one obtains an experimental value of $\Theta$ which should consistently equal the conditional values that our theory predicts.

In principle, the entropic contribution of electrons gives an additional $T^2$ correction owing to the band-structure modification by thermal expansion[16]. However, this correction to $F$ is typically of parts in $10^4$ up to the highest temperature that quantum oscillations are observable[8]. Frequency shifts with a $T^4$ dependence have been observed for a few, non-magnetic metals; these shifts were attributed to the lattice contribution to thermal expansion[17,18], as well as the electron-phonon coupling[19] [see supplementary note 2(A)]. Because of their distinct power law ($T^4$) compared to the topological and Sommerfeld corrections ($T^2$), lattice and electron-phonon effects would in principle be easy to detect and analyze separately.

Next, we discuss the identification of band topology. Our approach senses the linearity of bands and hence the topological character needs to be inferred. As the argument is based on a $\mathbf{k} \cdot \mathbf{p}$ expansion, it is only applicable to small Fermi surfaces that are much smaller than the Brillouin zone. As topological materials of practical relevance generally host small Fermi pockets, this criterion is typically fulfilled. Even when linear bands with a small Fermi pocket are detected, a question remains about distinguishing massive from massless Dirac materials. Conservatively speaking, one can never completely rule out the possibility that any hypothesized Dirac fermion has a tiny mass $m_D$ which leads to a weakly nonlinear energy dispersion $E(k) = \pm\sqrt{(v_x \hbar k_x)^2 + (v_y \hbar k_y)^2 + m_D^2}$ and $\Theta = (1/16)|E_F|(|E_F| + 2|m_D|)/(|E_F| + |m_D|)^2$. The experimental uncertainty in $\Theta$ can be used to set an upper bound on $|m_D|$. If the dispersion at the Fermi level is experimentally indistinguishable from a linear one, a hypothetical gap at the node is necessarily much smaller than the chemical potential (measured from the node), and has negligible influence on the low-energy excitations. For example, graphene with a typical chemical potential (~meV) is well described by massless Dirac fermions, despite the existence of a spin-orbit-induced gap (~$\mu$eV)[20].

High temperatures are natural opponents of quantum oscillations, because discontinuous changes in the occupation of Landau levels are smoothened out by the Fermi-Dirac distribution, as illustrated in the top panel of Fig. 2. The characteristic $T^2$ dependence for $F$ is observable at a temperature scale $T^*$ where $2\pi^2 k_B T^*$ is comparable to the cyclotron energy, while for $T \gg T^*$ oscillations are exponentially suppressed by thermal damping[8]. The optimal temperature window

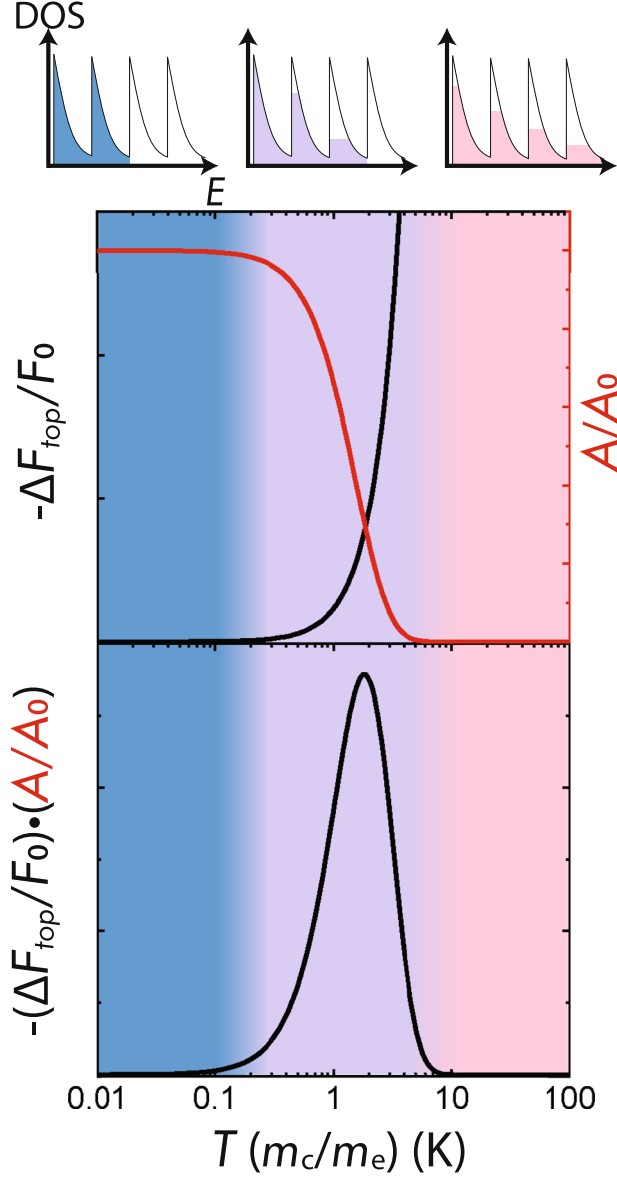

**Fig. 2 Optimal temperature range for detecting topological frequency shift.** The upper square panel displays the temperature dependence of the quantum-oscillation amplitude ($\Delta A = A(T) - A_0$) and topological frequency shift $\Delta F_{top}(T)$. The temperature axis is scaled by the ratio of cyclotron to free-electron mass, and $F_0$ and $A_0$ stand for the frequency and amplitude at zero temperature, respectively. $-\Delta F_{top}/F_0$ steeply increases just before the oscillation amplitude vanishes; this corresponds to the temperature regime where thermal broadening is comparable to the cyclotron energy ($\varepsilon_c \approx 2\pi^2 k_B T$), as illustrated by the filled density-of-states (DOS) plot for various temperatures (at the very top of figure). The lower square panel plots the temperature dependence of $-(\Delta F_{top}/F_0) \cdot (A/A_0)$; its peak identifies the optimal temperature to observe the topological frequency shift; our scaling of the temperature axis implies the optimal temperature is inversely proportional to the cyclotron mass.

for observing $F(T)$ is determined approximately by plotting the product (of the frequency shift and the amplitude) as a function of $T$, as in Fig. 2. Since $T^*$ is inversely proportional to the effective mass $m_c$, the requirement for elevated temperatures does not preclude their observation even when quasiparticles masses are heavy, it simply reduces the optimal temperature window to lower values. For example, for $m_c$ that is ten times the free-electron mass, Fig. 2 predicts the optimal temperature window to be between 0.1 and 1 K.

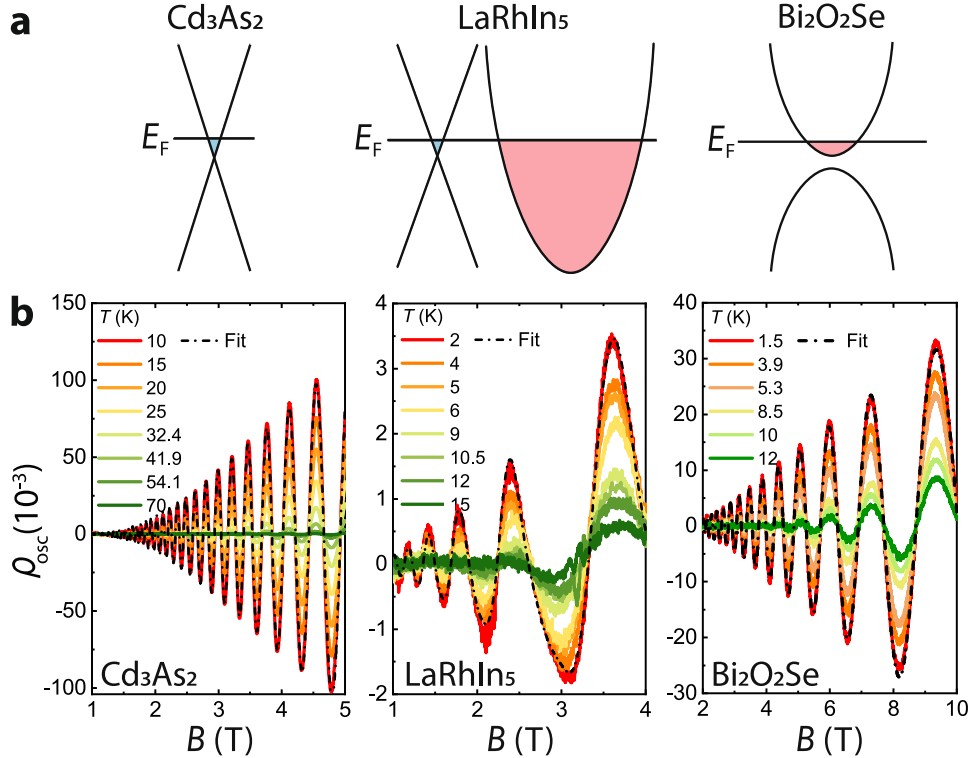

**Fig. 3 Experimental results of temperature-dependent quantum-oscillation measurements. a** Band-structure illustration of three different types of materials, including $Cd_3As_2$ and $Bi_2O_2Se$ where only one Fermi pocket and its symmetric copies sit at the Fermi level, as well as $LaRhIn_5$ where the small, candidate Dirac Fermi pocket coexists with large trivial pockets. **b** Temperature-dependent SdH oscillations of small Fermi pockets for $Cd_3As_2$, $LaRhIn_5$ and $Bi_2O_2Se$ respectively. $\rho_{osc} = \Delta\rho/\rho_{BG}$, with $\Delta\rho$ the oscillatory part of the magnetoresistivity, and $\rho_{BG}$ a polynomial fit to the smooth background. The dashed black line represents the Lifshitz–Kosevich fit to the quantum oscillation measured at lowest temperature for each material. The magnetic field range is chosen to include as many low-noise quantum oscillations as possible, to improve the accuracy of frequency fitting. Exact device geometry and field/current orientations are described in the supplementary Material.

Our method hence is expected to apply to strongly interacting topological materials with strong mass renormalization. Detailed discussion on applicability to heavy-fermion materials can be found in supplementary note 2(C).

**Detection and analysis of temperature-dependent quantum-oscillation frequency.** Experimentally, these predictions turn out to be readily observable. Three distinct materials were analyzed (Fig. 3): (i) $Cd_3As_2$ is a well-studied prototypical Dirac semimetal with a time-reversal-related pair of Dirac-type Fermi pockets and no other pockets ($F \approx 43.7$ T)[21]. (ii) $Bi_2O_2Se$ is a topologically trivial semimetal with a single, small electron pocket centered at $\Gamma$ ($F \approx 33.3$ T)[22]. (iii) $LaRhIn_5$ is a large-carrier-density, multiband metal that hosts Brillouin-zone-sized pockets[23] in addition to a very small pocket ($F \approx 6.9$ T) that is inconsistent with conventional Schrödinger-like behavior[24]. In their pioneering work, Mikitik and Sharlai proposed that the small pocket encloses a Dirac nodal line[25], based on an assumption that the spin-orbit coupling is perturbatively weak. However, our first-principles calculation [detailed in Supplementary note 2(D) and supplementary note 4] suggest this assumption to be unjustified, leaving the topology of the small pocket still in question.

Crystalline microbars of (i–iii) for four-terminal resistivity measurements were prepared by Focused Ion Beam machining[26]. These microbars feature optimized geometries for longitudinal transport and provide high signal amplitudes even in the highly conductive materials studied here. All samples show pronounced quantum oscillations of the longitudinal magnetoresistance, with a single small frequency in the low-field regime (Fig. 3). We

obtain $F_0(E_F)$ from extrapolating the temperature-dependent oscillation frequency to zero temperature, and $m_c$ from the temperature dependence of the amplitude. For each material, the temperature dependence of the frequency is obtained from fitting the entire experimental dataset [measured at different temperatures and fields (see Fig. 4)] to a single Lifshitz–Kosevich formula[9]. The fitting is done by a standard least squares regression method using the nonlinear model fitting function provided by Mathematica, as detailed in supplementary note 3(A) and (B).

Both $Cd_3As_2$ and $Bi_2O_2Se$ are low-carrier-density materials in which the Sommerfeld correction applies, and accordingly a clear frequency shift, $\Delta F(T)$, is observed in both of them (Fig. 4). It is evident from the raw data that $Cd_3As_2$ exhibits a stronger frequency shift compared to the trivial $Bi_2O_2Se$. For both cases, $\Delta F(T)$ falls directly onto the theoretical predictions $\Theta(\pi k_B T)^2/\beta^2 F_0(E_F)$ from Eq. (3) for the trivial and topological case respectively. For $Cd_3As_2$, $\Theta = 5/48$, as predicted for a Dirac semimetal with only two time-reversal-related Fermi pockets; for $Bi_2O_2Se$, $\Theta = 1/48$ is consistent with a conventional semimetal with a single Fermi pocket. We emphasize that the coefficient of $T^2$ has no tuning parameter as $F_0(E_F)$ and $m_c$ are fixed by measurement results, i.e., theory fixes $\Theta$ to take on different rational values depending on whether the pocket is Schrödinger or Dirac type.

In comparison, $LaRhIn_5$ is a high-carrier-density metal, and hence the measured $\Delta F(T)$ (for three devices) directly matches the prediction of $\Theta = 1/16$, a purely topological correction. This confirms the small 7 T pocket of $LaRhIn_5$ is Dirac type, with a chemical potential that is pinned by other large coexisting pockets.

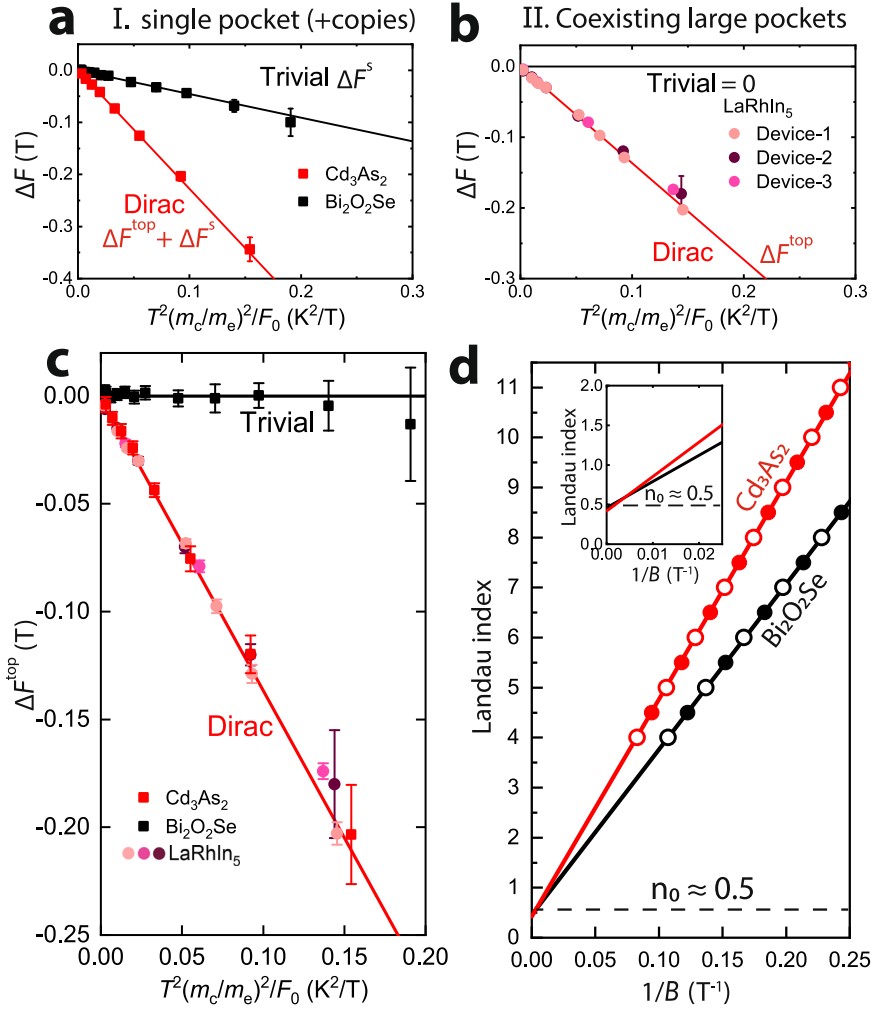

**Fig. 4 Analysis of frequency shift and Landau-fan plots. a** The total frequency shift $\Delta F(T)$ versus $T^2(m_c/m_e)/F_0$ for $Cd_3As_2$ and $Bi_2O_2Se$, both of which are semimetals with a single Fermi pocket plus symmetry-related copies (if any). **b** $\Delta F(T)$ for $LaRhIn_5$, which has a single, small pocket with coexisting large pockets. The solid lines display the fitting-parameter-free theoretical expectations $\Theta(\pi k_B T)^2/\beta^2 F_0$. For low-carrier density semimetal, $\Theta = 5/48$ for Dirac case while for trivial case $\Theta = 1/48$. For Dirac metal with high-carrier density $\Theta = 1/16$. The filled squares and circles represent the experimentally determined value for different materials. **c** The frequency shift after subtracting the Sommerfeld contribution ($\Delta F^s$); what remains is the topological frequency shift, if any. $\Delta F - \Delta F^s$ for both $LaRhIn_5$ (circles) and $Cd_3As_2$ (red squares) fall on the expected theoretical line (solid, red) for Dirac semimetal/ metal, while for the topologically trivial semimetal $Bi_2O_2Se$, the frequency shift nearly vanishes after subtraction. **d** Landau-fan plots of $Cd_3As_2$ and $Bi_2O_2Se$. The solid and empty symbols represent valleys and peaks of the quantum oscillations. Following the common interpretation of the residual Landau index ($n_0$ times $2\pi$) as a quantized Berry phase, one would erroneously diagnose both $Cd_3As_2$ and $Bi_2O_2Se$ as topologically nontrivial. All error bars in this figure are determined by the standard error of the fitting parameters generated by the nonlinear regression fitting procedure.

The universal topological aspect of these distinct compounds becomes evident after subtracting the Sommerfeld correction described by supplementary Eq. (10) and (11) from the experimental $\Delta F$ for $Bi_2O_2Se$ and $Cd_3As_2$. Remarkably, $Cd_3As_2$ and all three devices of $LaRhIn_5$ collapse on the same red line in Fig. 4c despite their highly different band structures and microscopic details, highlighting the common topological origin of $\Delta F^{top}$ and its insensitivity to material-specific details. Instructively, a quantum-oscillation phase analysis of $Bi_2O_2Se$ and $Cd_3As_2$ by the Landau-fan-diagram method uncovers a $\pi$-phase shift in both of them (see Fig. 4d), despite their clearly distinct topology, exemplifying the faults of this method.

These results can be further quantitatively strengthened by the self-consistency of $|E_F|$ computed by two different ways. From $k \cdot p$ theory, it can be expressed in terms of the standard Lifshitz–Kosevich parameters: $|E_F| = 2e\hbar F_0/m_c$ for the linearized Dirac pocket, and $|E_F| = e\hbar F_0/m_c$ for the quadratic Schrödinger pocket. On the other hand, $|E_F|$ can also be determined via $\Delta F(T)$:

in the Dirac case, $|E_F| = |\frac{\partial E}{\partial(\log m_c)}|$ [cf. Eq. (1)] is deducible from the topological correction, while in the Schrödinger case $|E_F|$ is deducible[15] from the Sommerfeld correction to $F$. For the model correctly describing the topology of the pocket, both estimates for $|E_F|$ should be consistent, and indeed this is what Table 1 shows. This allows for a simple self-consistency check. By analyzing a measured temperature-dependent quantum-oscillation frequency $F(T)$ within both the Dirac/Weyl and Schrödinger framework, the match of both $|E_F|$ signals the correct $k \cdot p$ model.

## Discussions

To conclude, our experimental methodology allows us to diagnose the linear dispersion of small, topological Fermi pockets, as demonstrated by our three case studies. Despite their microscopic differences, all Dirac/Weyl/multifold fermions have a linear dispersion close to the nodal degeneracy. The linear dispersion is directly sensed by the temperature dependence of the oscillation

**Table 1 Self-consistency check for distinguishing topological or Schrödinger-type pocket. For each material, the Fermi energy (in units of meV) is derived from $\Delta F/T^2$ and $F_0$ [Eq. (1)] assuming the pocket is either Dirac or Schrödinger type [see supplementary note 2(B)]. Here $g$ stands for the zero-field density of states. The results clearly identify both $Cd_3As_2$ and $LaRhIn_5$ as topological materials, while for $Bi_2O_2Se$ it clearly reveals its topologically trivial nature.**

| Materials | Topological | | Schrödinger | |
|---|---|---|---|---|
| | $\|E_F\| = \left\|\frac{\partial E}{\partial(\log m_c)}\right\|$ | $\|E_F\| = 2e\hbar F_0/m_c$ | $\|E_F\| = (1/2)\left\|\frac{\partial E}{\partial(\log g)}\right\|$ | $\|E_F\| = e\hbar F_0/m_c$ |
| $Cd_3As_2$ | 278.9 | 273 | 55.8 | 136.5 |
| $LaRhIn_5$ | 23.7 | 24 | / | 12 |
| $Bi_2O_2Se$ | 201.9 | 42.8 | 20.2 | 21.4 |

frequency when the Fermi level is also close to the node. It is also capable of identifying multi-Weyl fermions protected by crystallographic rotational symmetry, e.g., the dispersion of a double–Weyl (triple-Weyl) fermion is quadratic (cubic) in two momentum directions and linear in the third direction[27], hence they would be identifiable by measuring $F(T)$ at various field orientations. Our methodology is applicable independent of the magnitude of the Zeeman splitting of Landau levels; this magnitude would affect the relative amplitudes of higher harmonics of quantum oscillations[9] but not their frequency.

Because the energy scale for band inversion tends to be small compared to the bandwidth, topological Fermi pockets are often small compared to the Brillouin zone. It is not impossible for highly inverted materials that the dispersion of larger topological Fermi pockets acquires substantial nonlinear corrections, in which case our methodology becomes less useful for topological diagnosis. It is also less straightforward to subtract the Sommerfeld correction (from the frequency shift) in materials where a topological Fermi pocket coexists with a non-topological pocket of comparable size; here, supplementing our methodology with a first-principles calculation may be useful to isolate the topological frequency shift.

It will be interesting to apply our method to investigate the topology of Fermi-liquid materials with extreme interaction-driven mass enhancement (e.g., heavy-fermion materials), with the caveat that the optimal temperature for measuring $F(T)$ is inversely proportional to the effective mass. The presence of Dirac fermions in $LaRhIn_5$ suggests that the intimately related $Ce(Co, Rh, Ir)In_5$ are prime candidates for such efforts. The Ce family has a similar band structure to $LaRhIn_5$, but are more prone to correlation-driven instabilities such as unconventional superconductivity[28]. A different class of correlated topological metals include the Kondo–Weyl semimetals[29,30], in which itinerant electronic bands hybridize with localized (4f/5f)-states leading to strongly renormalized Weyl dispersions.

As the field matures towards strongly interacting topological matter, it leaves the comfort zone in which ab-initio-predicted topological band structures can straightforwardly be confirmed by angle-resolved photoemission spectroscopy. The new conceptual and experimental challenges—arising from competing, near-degenerate ground states and unconventional superconductivity—call for new approaches to detect low-energy excitations and assess their topological character. Extending the framework of quantum oscillations to topologically nontrivial Fermi surfaces, as presented here, will play a major role in this development.

## Methods

**Microstructure fabrication**. Micro-devices of all materials are fabricated with a FEI Helios Plasma FIB using Xe-ions. Thin slab (lamella) was dig out from a single crystalline material, which was transferred and glued down to a sapphire substrate. The transferred lamella was later patterned to the desired geometry with the Plasma FIB.

**Band-structure calculations**. To check for the robustness of band-structure results against choices of functionals in the calculation, the calculations were performed via two independent methods which are both detailedly described in supplementary note 4(A).

## Data availability
Data that support the findings of this study are deposited to Zenodo with the access link: https://doi.org/10.5281/zenodo.5482689.

## Code availability
Mathematica code used for the Lifshitz–Kosevich fit of temperature-dependent quantum oscillations can be found via the access link: https://doi.org/10.5281/zenodo.5482689.

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

## Acknowledgements

A.A. was supported initially by the Yale Postdoctoral Prize Fellowship, and subsequently by the Gordon and Betty Moore Foundation EPiQS Initiative through Grant No. GBMF 4305 and GBMF 8691 at the University of Illinois. S.Z, Q.W. and O.V.Y. acknowledge support by NCCR Marvel and the Swiss National Supercomputing Centre (CSCS) under Projects No. s832 and No. s1008. M.D.B. acknowledges studentship funding from EPSRC under grant no EP/L015110/1. E.D.B. and F.R. were supported by the U.S. DOE, Basic Energy Sciences, Division of Materials Sciences and Engineering. C.P. and P.J.W.M. acknowledge funding by the European Research Council (ERC) under the European Union's Horizon 2020 research and innovation programme ("MiTopMat"—grant agreement No. 715730). This project was funded by the Swiss National Science Foundation (Grants No. PP00P2_176789).

## Author contributions

Crystals were synthesized and characterized by F.R., E.D.B. ($LaRhIn_5$), T.T., H.P. ($Bi_2O_2Se$) and N.K., C.S., C.F. ($Cd_3As_2$). The experiment design, FIB microstructuring and the magnetotransport measurements were performed by C.G., C.P., A.E., K.R.S., M.B., P.J.W.M. A.A. developed, applied and described the theoretical framework, and the analysis of experimental results has been done by C.G. and C.P. Band structures were calculated by F.F., S.Z., Q.W., O.Y., C.F., and Y.S. All authors were involved in writing the paper.

## Competing interests

The authors declare no competing interests.
