## [Peer Review File · Nature Communications]

REVIEWER COMMENTS

Reviewer #1 (Remarks to the Author):

In this manuscript, Guo et al. claimed that temperature dependence of oscillation frequency is also related to material topology and demonstrated results when applying the technique to well-known semimetal and metal. To be honest, however, I think the presented analysis does not have so large impact on this research field, and still, ab-initio calculations and advanced spectroscopies such as ARPES and STM are much more direct and certain for discovering topological materials. In particular, they also admit that "Our approach senses the linearity of bands and hence the topological character needs to be inferred". So I cannot recommend the publication in Nature Communications. I also have some questions about the presented results. I recommend the authors to address them before submitting it to a more specialized journal.

1. I agree with the authors that "Our approach senses the linearity of bands and hence the topological character needs to be inferred". On the other hand, they also claimed that "Graphene for example is well described by massless Dirac fermions, despite the existence of a gap in the μeV range due to spin-orbit interactions." However, this context sounds very misleading. The present analysis cannot determine whether the material has a gap or not even in much wider energy scales, say meV. A tiny gap of μeV (~ 10 mK) does not so matter even in ultralow temperature measurements, but a gap of meV (~ 10 K) matters. Title and statements in introduction and summary (such as that the new conceptual and experimental challenges – arising from competing, near-degenerate ground states and unconventional superconductivity – call for new approaches to detect low-energy excitations and assess their topological character) is too strong and not directly related to the achievement, since the topological character needs to be inferred in the present method. They need to be thoroughly revised.

2. Even in Cd₃As₂, its electronic structure is no so simple, unlike Fig. 3(a). First, there are two Dirac cones and obviously the Fermi level is above the saddle point of the two Dirac cones, considering that $F(T)$ is as large as 44 T (see, for example, Z. Wang et al., PRB (2013)). In such a case, the phase should be strongly dependent on the Fermi level position relative to the saddle point energy (see, for example, C. M. Wang et al., PRL (2016)), and actually the phase values are not ideally 0.5 and scattered in many previous reports.

Reviewer #2 (Remarks to the Author):

The authors of this manuscript proposed a new methodology to diagnose the band topology from the temperature dependence of quantum oscillation frequency. Their combined theoretical and experimental studies demonstrated the linear band dispersion in topological semimetals with a small Fermi pocket leads to a striking topological correction to the quantum oscillation frequency, resulting in a quadratic temperature dependence of the change of quantum oscillation frequency (Δ_F). Experimentally, the authors studied quantum oscillations of three representative materials, including a prototypical Dirac semimetal Cd₃As₂, a multiple band material LaRhIn₅ which features coexistence of a small Dirac pocket and a large trivial pocket, and a trivial semimetal Bi₂O₂Se with a small pocket. The authors performed careful magnetotransport measurements on these materials using microbar samples prepared through FIB cutting and obtained the temperature dependence of quantum oscillation frequency from the fits of the SdH quantum oscillation data by LK formula for various temperatures. After subtracting the Sommerfeld contribution, the authors found the change of quantum oscillation frequency Δ_F with temperature in Dirac semimetals (Cd₃As₂ and LaRhIn₅) can be precisely described by the theory that predicts the quadratic temperature dependence of

Δ_F is governed by the energy derivative of the cyclotron mass; by contrast, for the trivial semimetal Bi₂O₂Se, Δ_F vanishes after subtracting Sommerfeld contribution.

I think the work presented in this manuscript is of high quality and very interesting. The current approach for identifying band topology from quantum oscillation data is based on the Berry phase analyses using Landau level (LL) index fan diagram. However, there have been intensive debates over the definition of integer LL indices (minimal vs. maximal resistivity, or minimal vs. maximal conductivity). Moreover, as indicated by the authors, Berry phase can also be affected by other factors such as magnetism. I believe the new methodology demonstrated in this manuscript could complement the Berry phase analysis and thus has a potential to facilitate discoveries of novel topological materials based on quantum oscillations. I recommend this manuscript for publication in Nature Communications.

However, I find a number of issues (listed below) in the current version of manuscript, which should be addressed before acceptance.

1. On page 5 (lines 90-92), the authors stated "... For single-frequency, low-carrier-density semimetals, the Sommerfeld correction is of the same order of magnitude as the topological correction, giving $\theta = 1/48$ for parabolic bands and $\theta = 5/48$ for linear bands.." . I note θ is the coefficient of the frequency correction term in eq. 3. What are the values of θ for Cd₃As₂, LaRhIn₅ and Bi₂O₂Se? The authors did not mention it in the discussions of the data shown in Fig. 4.
2. How is the Sommerfeld contribution subtracted in Fig. 4c? This is not clearly described in the manuscript. Given that the Sommerfeld correction to F is the order of $(k_B T)^2 / \beta v_F$, how is the Sommerfeld correction term different among the three studied materials?
3. I feel the various sections of supplementary materials (SM) are not appropriately cited in the main text. Without reading the entire SM, it is difficult to understand the main text. Some of the important theoretical parts of the SM may be moved to the main text so that the reader can easily follow.
4. The fit of m_c should be presented.
5. How E_F is deduced from Sommerfeld correction to F for the Schrodinger case (Table 1).
6. How are the error bars in Fig. 4a-4c defined?
7. In the caption of Fig. 2, it states "the oscillation frequency strongly increases just before the quantum oscillation vanishes". This statement is misleading. I believe what the authors meant to say is that the relative change of the oscillation frequency steeply increases just before the oscillation vanishes.
8. Line 127 on page 7, "optical temperature" should be "optimal temperature".
9. Line 167, "see Fig. 4a" should be "see Fig. 4d".
10. On line 184 of SM, the authors stated substituting Eq. (4) of the main text to Eq. (S7).... I did not find Eq. 4 in the main text.
11. In Fig. S8, why are the error bars large at low temperatures, but small/absent at high temperatures where quantum oscillations are strongly suppressed.
12. Fig. S8 show $\Delta F(T)$ is 0.1-0.3T at 15K, but the largest ΔF shown in Fig. 4a reaches 0.5T. They do not seem consistent.
13. Table S1 lists the fitting parameters obtained from the fit of the SdH data by Eq. S13. It is difficult for me to understand why $dF/d(T^2)$ is a fitting parameter.

Reviewer #3 (Remarks to the Author):

Chunyu Guo and coauthors proposed a new experimental methodology to estimate the topological nature of the materials from the temperature-dependent quantum oscillations. The effective mass of

electrons varies with Fermi energy in the linear band while keeping the same in the parabolic band, leading to the different frequency shift when considering T^2 -correction to the traditional Lifshitz-Kosevich formula, which can distinguish topological contribution from the parabolic one after calculating the T^2 related frequency shift. In three representative materials, the self-consistency new method becomes a powerful tool, which greatly replenishes the conventional limited Berry phase-shift method. I think the manuscript is well-organized and may attract wide interest, but the authors should address the following comments or questions before I recommend the acceptance of the paper:

1. Other possibilities of temperature-dependent frequency shift need to be excluded very clearly, such as impurity or disorder-related extrinsic origin. Also, the discussion about electron-phonon interaction in SI should be added to the main text and clarify it more persuasively, for the temperature at which the oscillations disappear is high enough, even 70K in Cd₃As₂, to cause strong phonon scattering.
2. While the manuscript cited the ref [27], the reasonable explanation why the correction to the Lifshitz-Kosevich formula is T^2 related is needed.
3. The quasiparticle Dirac/Weyl fermion with the linear band is a typically collective excitation mode, the hypothesized Dirac fermion with a tiny mass m_D certainly leads to a weakly-nonlinear energy dispersion, while this fact doesn't affect the topology of the band, the discussion at lines 110-116 is redundant for the main point of the new method to detect non-trivial linear band from the trivial parabolic band.
4. The authors didn't mention the top panel of figure 2 in the main text.
5. In figure 3(b), the unit of the y-axis should be shown.
6. The authors need to explain the different magnetic field range in three different type samples in Figure 3(b).
7. The frequency shift is as small as 0.8% in Cd₃As₂, so the accurate fitting of frequency is significantly important in this new method, the random error of the conventional calculation method is much larger than the resolution in this manuscript. A more detailed fitting process is needed, especially about the variance and error bars.
8. The Cd₃As₂ is generalized considered a Dirac semimetal with a single Dirac cone, however, the frequency shift versus the normalized T^2 in figure 4 (a-b) showed two components in Cd₃As₂, while they showed only one in LaRhIn₅, in which a linear band and a parabolic band contribute to the quantum oscillation, please explain it.
9. The applicability and limitation of this new method should be summarized in the last part.

Dear Reviewers,

We thank you for taking the time to carefully review our manuscript “Fingerprint of topology in quantum oscillations at elevated temperatures”. Your expressed interest and expected impact on the community is highly appreciated, and indeed it is gratifying to see that already our arXiv version has inspired the first papers appear that employ this new method (T. Terashima et al., arXiv: 2105.14733 (2021)). At the same time, we equally appreciate your pointed comments that helped us to sharpen the main message and render the work both more rigorous and more accessible.

Thank you again for your efforts and support. In the following, we will address the comments on a point-by-point basis:

REVIEWER COMMENTS

Reviewer #1 (Remarks to the Author):

In this manuscript, Guo et al. claimed that temperature dependence of oscillation frequency is also related to material topology and demonstrated results when applying the technique to well-known semimetal and metal. To be honest, however, I think the presented analysis does not have so large impact on this research field, and still, ab-initio calculations and advanced spectroscopies such as ARPES and STM are much more direct and certain for discovering topological materials. In particular, they also admit that “Our approach senses the linearity of bands and hence the topological character needs to be inferred”. So I cannot recommend the publication in Nature Communications. I also have some questions about the presented results. I recommend the authors to address them before submitting it to a more specialized journal.

Thank you for your candid assessment. Let us here present a clear counterargument to this line of thought.

1) Quantum oscillations have been a key driver of the field of topological semimetals. Without any doubt, quantum oscillations are already much more

commonly used to make statements about a materials topological character than STM and ARPES combined. A quick search on WebOfScience finds: “Weyl semimetal” + “quantum oscillations” 231; “Weyl semimetal” + “ARPES” 60; “Weyl semimetal” + “STM” 15. This central role of quantum oscillations is no surprise. While ARPES and STM require either large-scale facilities or highly specialized laboratories, magnetic quantum oscillations are readily observable in semi-metals in wide-spread laboratory magnets. Most unfortunately, the framework in which these quantum oscillations were analyzed, via a Landau fan diagram, is not rigorous in many symmetry classes. Our approach here allows to draw much more stringent conclusions, and hopefully will replace this analysis. This is already happening as other groups publish results based on our analysis, e.g. T. Terashima et al., arXiv: 2105.14733 (2021).

- 2) Quantum oscillations can probe bulk topology without ab-initio calculations. Often, ARPES and quantum oscillations are discussed as competition for the best method to determine electronic spectra (surface sensitivity, need for high field, energy resolution, etc.). In reality, the best results are obtained when they are combined. However, it is evident that as the field is turning towards strongly correlated and magnetic topological materials, it leaves the comfort zone of ARPES. Once strong electronic interactions renormalize the band structure, ARPES becomes inapplicable. The temperature scales under strong photon irradiation preclude any reliable measurements below 10 K in almost all beamlines, which greatly exceeds interaction-induced phases such as Kondo temperatures. Even where temperature is not a concern, ARPES famously lacks the energy resolution to obtain quasiparticle effective masses in interacting scenarios. With great efforts, the el-ph renormalization in conventional superconductors can nowadays be resolved; yet this is trivial in a PPMS-type system operating at 2 K = 170 μ eV. STM certainly remains competitive, yet the interpretation of QPI/tunneling data is extremely model dependent – and a detailed model of the electronic structure is a luxury that is not expected in correlated topological materials.

This list could easily be expanded and the advantages and disadvantages of the various techniques listed. At the end, this is not our point. All these methods, in combination with electronic modeling, are required to fully understand a material.

We do not see this as a replacement to ARPES or STM, but as a complementary method.

1. I agree with the authors that “Our approach senses the linearity of bands and hence the topological character needs to be inferred”. On the other hand, they also claimed that “Graphene for example is well described by massless Dirac fermions, despite the existence of a gap in the μeV range due to spin-orbit interactions.” However, this context sounds very misleading. The present analysis cannot determine whether the material has a gap or not even in much wider energy scales, say meV. A tiny gap of μeV (~ 10 mK) does not so matter even in ultralow temperature measurements, but a gap of meV (~ 10 K) matters.

Thank you for pointing out a confusing issue. What we should have written is that “For example, graphene with a typical chemical potential ($\sim \text{meV}$) is well described by massless Dirac fermions, despite the existence of a spin-orbit-induced gap ($\sim \mu\text{eV}$).”

The corrected sentence now clarifies that the massless-Dirac description holds if the gap is much smaller than the chemical potential. The reason, of course, is that the kinetic energies of these Dirac fermions so massively exceed their rest mass.

The exact same argument holds for our experimental case studies, where the chemical potential is high enough (10’s and 100’s of meV) that a 1 meV gap (if it exists) would not cause a significant deviation from a linear dispersion. Even at the highest temperatures that oscillations were measured (50 K = 4.3 meV for Cd_3As_2), we have found no deviation from a linear dispersion.

We have elaborated on this issue in the main text.

Title and statements in introduction and summary (such as that the new conceptual and experimental challenges – arising from competing, near-degenerate ground states and unconventional superconductivity – call for new approaches to detect low-energy excitations and assess their topological character) is too strong and not directly related to the achievement, since the topological character needs to be inferred in the present method. They need to be thoroughly revised.

We disagree with the referee that the title “Fingerprint of topology...” and related claims in the Concluding section are too strong just because topology must be “inferred”. In fact, we show the inference is well justified for small Fermi pockets – a point we have emphasized throughout this work.

We have added to the revised manuscript a paragraph in the Conclusion section that discusses the limitations of our methodology, which directly states that our methodology is less useful for large topological pockets.

2. Even in Cd_3As_2 , its electronic structure is not so simple, unlike Fig. 3(a). First, there are two Dirac cones and obviously the Fermi level is above the saddle point of the two Dirac cones, considering that $F(T)$ is as large as 44 T (see, for example, Z. Wang et al., PRB (2013)). In such a case, the phase should be strongly dependent on the Fermi level position relative to the saddle point energy (see, for example, C. M. Wang et al., PRL (2016)), and actually the phase values are not ideally 0.5 and scattered in many previous reports.

We agree that there is a lot of confusion about the electronic structure of Cd_3As_2 in the community. However, the referee’s claim that the experimental Fermi level is above the saddlepoint (based on comparing our oscillation frequency of $F = 44\text{T}$ with a first-principles calculation by Z. Wang et al., PRB (2013)) is factually incorrect.

- 1) **Our quantum oscillations self-consistently exclude that the chemical potential is above the saddle point.** If that were the case, a Lifshitz transition strongly changes the Fermi surface topology. Below the saddle point, two separate Dirac surfaces exist, which merge into one, dumbbell-shaped object at energies above the saddle point. This necessitates the observation of multiple, and strongly angle-dependent, frequencies through the appearance of a neck- and belly-frequency. This is not observed. In fact, the angle dependence clearly shows that the Fermi

[Redacted]

surfaces are almost perfectly spherical. There have been reports of Cd_3As_2 above the Lifshitz transition, but note that in these cases an angle-dependence of frequencies is indeed observed [e.g. Y. Zhao et al., PRX 5,031037 (2015)]

- 2) **The electronic structure is far from definitely understood.** While the paper by Wang et al. was a pioneering work that inspired our experimental interest in the compound, it far from settles the question of the electronic structure. It is important to state that their work aimed to show a topological property of Cd_3As_2 being a Dirac semi-metal, not to capture the details of the band structure at high precision. These computations are very challenging, as they are multi-band systems with the important features in the meV range. Given the difficulty that GGA has to determine band gaps, it is not surprising that the theoretical values in literature for the gap parameter E_D vary by an order of magnitude.

[Redacted]

Here we reproduce an overview from [I. Crassee et al., Phys. Rev. Mat. 2, 120302 (2018)] that shows some selected cases of the published values of E_D . Not surprisingly, all agree it is a Dirac semi-metal (by symmetry) while the detailed numerics are different. Given the significant spread of E_D also in the experimental works, one may envision subtle differences in the crystal structure between different growths to modify the band structure details around the Dirac node. This may be related to the (ongoing) debate about the true crystal structure which has been proposed to require enormous unit cells due to vacancy ordering (e.g. 160 atom unit cells in M.N. Ali et al., Inorg. Chem. 53, 4062 (2014)). In light of the massive spread of both experimental and theoretical values for the saddle point, the fact that both the angle-dependence and the temperature-dependence

observed in our crystals clearly demonstrate a Dirac dispersion appears perfectly consistent with published literature.

3. Even within the calculation by Wang we would be below the saddle point.

While we stress that one should not overemphasize the details of the calculations by Wang et al., a quick check shows that even within this calculation our Fermi surface would be below the saddle point given a frequency of 44 T.

Using the parameters from Wang et al., we computed the expected quantum oscillation frequencies and the saddle point position. At $F = 44$ T, we would be well below the saddle point and hence within the regime of two separate Dirac pockets. Please note the extreme angle dependence of the frequencies above the saddle point, while below the calculation predicts an almost angle-independent frequency – in perfect agreement with our data.

Reviewer #2 (Remarks to the Author):

The authors of this manuscript proposed a new methodology to diagnose the band topology from the temperature dependence of quantum oscillation frequency. Their combined theoretical and experimental studies demonstrated the linear band dispersion in topological semimetals with a small Fermi pocket leads to a striking topological correction to the quantum oscillation frequency, resulting in a quadratic temperature dependence of the change of quantum oscillation frequency (Δ_F). Experimentally, the authors studied quantum oscillations of three representative materials, including a prototypical Dirac semimetal Cd_3As_2 , a multiple band material LaRhIn_5 which features coexistence of a small Dirac pocket and a large trivial pocket, and a trivial semimetal $\text{Bi}_2\text{O}_2\text{Se}$ with a small pocket. The authors performed careful magnetotransport measurements on these materials using microbar samples prepared through FIB cutting and obtained the temperature dependence of quantum oscillation frequency from the fits of the SdH quantum oscillation data by LK formula for various temperatures. After subtracting the Sommerfeld contribution, the authors

found the change of quantum oscillation frequency Δ_F with temperature in Dirac semimetals (Cd_3As_2 and LaRhIn_5) can be precisely described by the theory that predicts the quadratic temperature dependence of Δ_F is governed by the energy derivative of the cyclotron mass; by contrast, for the trivial semimetal $\text{Bi}_2\text{O}_2\text{Se}$, Δ_F vanishes after subtracting Sommerfeld contribution.

I think the work presented in this manuscript is of high quality and very interesting. The current approach for identifying band topology from quantum oscillation data is based on the Berry phase analyses using Landau level (LL) index fan diagram. However, there have been intensive debates over the definition of integer LL indices (minimal vs. maximal resistivity, or minimal vs. maximal conductivity). Moreover, as indicated by the authors, Berry phase can also be affected by other factors such as magnetism. I believe the new methodology demonstrated in this manuscript could complement the Berry phase analysis and thus has a potential to facilitate discoveries of novel topological materials based on quantum oscillations. I recommend this manuscript for publication in Nature Communications.

However, I find a number of issues (listed below) in the current version of manuscript, which should be addressed before acceptance.

1. On page 5 (lines 90-92), the authors stated "... For single-frequency, low-carrier-density semimetals, the Sommerfeld correction is of the same order of magnitude as the topological correction, giving $\theta = 1/48$ for parabolic bands and $\theta = 5/48$ for linear bands..". I note θ is the coefficient of the frequency correction term in eq. 3. What are the values of θ for Cd_3As_2 , LaRhIn_5 and $\text{Bi}_2\text{O}_2\text{Se}$? The authors did not mention it in the discussions of the data shown in Fig. 4.

Thank you for pointing this out. We've now added the value of θ for all materials in the revised manuscript and explicitly into the caption of Fig. 4, which are listed below as well:

$\theta = 1/16$ for LaRhIn_5 , small Dirac pocket in a high carrier density metal.

$\theta = 1/48$ for $\text{Bi}_2\text{O}_2\text{Se}$, single-frequency, low-carrier-density trivial semimetal.

$\Theta = 5/48$ for Cd_3As_2 , single-frequency, low-carrier-density Dirac semimetal.

2. How is the Sommerfeld contribution subtracted in Fig. 4c? This is not clearly described in the manuscript.

We thank the referee for the comment. We've added two references to equations in the Supplemental Material, which describe exactly what is being subtracted.

Given that the Sommerfeld correction to F is the order of $(k_B T)^2 / \beta E_{\text{bw}}$, how is the Sommerfeld correction term different among the three studied materials?

A minor correction to the referee's claim is that *assuming the presence of Brillouin-zone-sized pockets*, the Sommerfeld correction to F is of the order of $(k_B T)^2 / \beta E_{\text{bw}}$, with β the effective Bohr magneton and E_{bw} the typical bandwidth. Ultimately, this follows from the temperature dependence of the chemical potential at fixed carrier density:

$$\mu(T) = E_F - \frac{1}{6} (\pi k_B T)^2 \frac{g'(E_F)}{g(E_F)} + O(T^4)$$

with g the zero-field density of states. The estimate $g'/g \sim E_{\text{bw}}$ holds only for Brillouin-zone sized pockets. This is the case for LaRhIn_5 , a multiband metal with high carrier density. The chemical potential is pinned by the wide bands with large number of carriers, which implies $(k_B T)^2 / \beta E_{\text{bw}}$ is much smaller than F (at zero temperature), and therefore the Sommerfeld correction in LaRhIn_5 is negligible.

For single frequency low carrier density semimetal such as $\text{Bi}_2\text{O}_2\text{Se}$ and Cd_3As_2 , $g'/g \sim E_{\text{bw}}$ does not hold, and in fact can be calculated exactly for 3D parabolic and linear bands.

For 3D parabolic bands, since g is proportional to \sqrt{E} , $\frac{g'(E_F)}{g(E_F)} \sim \frac{1}{2} \frac{1}{E_F}$. By replacing E_F with $E_F = e\hbar F_0 / m_c$, one obtains:

$$F(\mu, T) = F_0 - \frac{1}{48} \frac{(\pi k_B T)^2}{\beta^2 F_0} + O(T^4).$$

This is the Sommerfeld correction for $\text{Bi}_2\text{O}_2\text{Se}$.

On the other hand, for a 3D linear dispersive band, $g \sim E^2$, therefore $\frac{g'(E_F)}{g(E_F)} \sim 2 \frac{1}{E_F}$. Similarly by replacing E_F with $E_F = e\hbar F_0/m_c$ (the factor-of-two difference is due to the linear energy dispersion of momentum), the temperature dependence of frequency can be expressed as:

$$F(\mu, T) = F_0 - \frac{1}{24} \frac{(\pi k_B T)^2}{\beta^2 F_0} + O(T^4)$$

This is the Sommerfeld correction for the single frequency Dirac semimetal Cd_3As_2 .

To clearly address this point in the revised manuscript, we've extended our discussion on Sommerfeld correction in supplement Sec. II B, and clearly cited that in the main manuscript.

3. I feel the various sections of supplementary materials (SM) are not appropriately cited in the main text. Without reading the entire SM, it is difficult to understand the main text. Some of the important theoretical parts of the SM may be moved to the main text so that the reader can easily follow.

We thank the referee for the valuable suggestion. Indeed this point was discussed among us quite intensively, and we are very open to improving the format. Our issue was the following: We introduce a new idea conceptually and support it at the same time with experimental data. Such hybrid theory/experiment papers are always difficult to organize properly. We want to convey the main concept without being overly technical, yet we need to show that level of detail to demonstrate that this analysis is rigorous. We therefore adopted the strategy to write an extensive supplemental part, which almost constitutes a theory paper of its own. Then we focus on the main concepts and physical ideas in the main text and refer to the supplement for the detailed treatments.

We kept this strategy, and increased the citations of the supplement in the main text, to improve the link and clarity. We look forward to your comment if this was successful.

4. The fit of m_c should be presented.

Fig.R2 Cyclotron mass determination via Lifshitz-Kosevich fitting.

The experimental dataset measured at different temperatures and B fields is fitted globally by a standard least squares regression method using the non-linear model fitting function provided by Mathematica, with the effective mass as a fitting parameter; this is elaborated in Sec. III A of the supplement.

The above plot also shows the more traditional way of obtaining effective mass: by taking the peak in fast-Fourier-transform spectrum and fit its temperature dependence with Lifshitz-Kosevich theory. The two ways of obtaining effective masses are consistent with each other. We have added the above plot to Sec. III B of the SI.

5. How E_F is deduced from Sommerfeld correction to F for the Schrodinger case (Table 1).

In the Schrodinger case, $E_F = (1/2) g'/g$, with g the zero field density of states [cf. our reply to point 2]. In turn g'/g is extracted from the T^2 dependence of F , as detailed in Eq. S10 in supplement Sec. II B.

We believe the issue was the lack of a clear statement in the previous manuscript that D in Table 1 stands for the zero-field density of states, and a mismatch of notation (for the density of states) between main text and supplemental material. We've clarified that in the table caption of the revised manuscript, and also replaced D with g in Table 1 to streamline the notation. Note also that Sec II B of supplemental material is referred to in the caption of Table 1.

6. How are the error bars in Fig. 4a-4c defined?

As stated in the supplement Sec. III A, the entire experimental dataset measured at different temperatures is then fitted globally by a standard least squares regression method using the non-linear model fitting function provided by Mathematica. In this procedure, both the cyclotron mass m_c and the quantum mean free path l_q are set to be temperature-independent while the oscillation frequency F is not restricted to the same value at different temperatures, which directly yields the best-fit $\Delta F(T)$ to all experimental data. ***The error bar is determined by the standard error of the fitting parameters generated by the non-linear regression fitting procedure.***

In the revised manuscript, we now give a summary of the fitting procedure in the main text, and direct the interested reader to Supplemental Sec III for more details on the fitting methodology and error bar determination.

7. In the caption of Fig. 2, it states “the oscillation frequency strongly increases just before the quantum oscillation vanishes”. This statement is misleading. I believe what the authors meant to say is that the relative change of the oscillation frequency steeply increases just before the oscillation vanishes.

8. Line 127 on page 7, “optical temperature” should be “optimal temperature”.

9. Line 167, “see Fig. 4a” should be “see Fig. 4d”.

10. On line 184 of SM, the authors stated substituting Eq. (4) of the main text to Eq. (S7)... I did not find Eq. 4 in the main text.

We thank the referee for the detailed comments/corrections. These points are addressed in the revised manuscript.

11. In Fig. S8, why are the error bars large at low temperatures, but small/absent at high temperatures where quantum oscillations are strongly suppressed.

Thank you for the comment. This seeming difference is caused by the log-log manner of Fig.S8. Therefore although at low temperatures the error bar appears larger, it's actually much smaller, as shown in the figure below. As the log-log plot

is important for us to display the T^2 -dependence of frequency we will keep Fig. S8 in the log-log form but clarified this in the legend.

12. Fig. S8 show $\Delta F(T)$ is $0.1-0.3T$ at 15K, but the largest ΔF shown in Fig. 4a reaches $0.5T$. They do not seem consistent.

Here we combined Fig.S8 and Fig.4, one can see that the ΔF value is exactly the same, as these plots are using the same trace of data. We added a legend to the log-log plot and changed the scale of Fig.4.

13. Table S1 lists the fitting parameters obtained from the fit of the SdH data by Eq. S13. It is difficult for me to understand why $dF/d(T^2)$ is a fitting parameter.

As described previously, in our analysis the fitting parameter yields a temperature-dependent frequency $F(T)$. After that we fit the temperature dependence of $F(T)$ with $F(T)=F_0+AT^2$, therefore here parameter A [$dF/d(T^2)$] is produced by the polynomial fitting which makes it a fitting parameter. We've added detailed explanation to the caption of Table S1.

Reviewer #3 (Remarks to the Author):

Chunyu Guo and coauthors proposed a new experimental methodology to estimate the topological nature of the materials from the temperature-dependent quantum oscillations. The effective mass of electrons varies with Fermi energy in the linear band while keeping the same in the parabolic band, leading to the different frequency shift when considering T^2 -correction to the traditional Lifshitz-Kosevich formula, which can distinguish topological contribution from the parabolic one after calculating the T^2 related frequency shift. In three representative materials, the self-consistency new method becomes a powerful tool, which greatly replenishes the conventional limited Berry phase-shift method. I think the manuscript is well-organized and may attract wide interest, but the authors should address the following comments or questions before I recommend the acceptance of the paper:

1. Other possibilities of temperature-dependent frequency shift need to be excluded very clearly, such as impurity or disorder-related extrinsic origin. Also, the discussion about electron-phonon interaction in SI should be added to the main text and clarify it more persuasively,

for the temperature at which the oscillations disappear is high enough, even 70K in Cd_3As_2 , to cause strong phonon scattering.

We thank the referee for the detailed suggestion, the influence of electron-phonon interaction is now addressed in the revised manuscript with a clear reference to the detailed discussion in supplement. We remark that the electron-phonon interaction results in a T^4 -correction to the frequency, which would in principle be clearly distinguishable from the T^2 -correction that is the main subject of this work. In practice, this T^4 -correction is negligibly small in most materials, including Cd_3As_2 at $T=70\text{K}$ (we refer the referee to Fig.S8 which shows that the frequency correction can only be meaningfully fit to T^2).

We are not aware that disorder would result in a temperature-dependent frequency shift. We refer the referee to Sec 5.5 in Shoenberg's book "Magnetic oscillations in metals", which gives an authoritative summary of all empirically-observed frequency shifts (and their interpretations) for a wide range of metals. In Lifshitz-Kosevich theory, disorder would modify the oscillation amplitude and not the frequency. Though disorder is treated rather simplistically in Lifshitz-

Kosevich theory, such treatment seems to be well-justified empirically, e.g., the Lifshitz-Kosevich formula gives an excellent fit to all experimental data we have presented.

2. While the manuscript cited the ref [27], the reasonable explanation why the correction to the Lifshitz-Kosevich formula is T^2 related is needed.

It is useful to view the Lifshitz-Kosevich formula as an asymptotic expansion in powers of $k_B T/E_F$ (E_F being Fermi energy), with odd powers of T modifying the oscillation amplitude, and even powers of T modifying the frequency and phase. The first odd power of T gives the known thermal damping factor. We have merely extended the theory to the first even power of T , which affects the frequency. (These remarks are now added to the main text, to further motivate the T^2 correction.) The formal derivation is given in Sec. II A of the Supplemental Material.

For a physically intuitive explanation of the T^2 correction to frequency: due to the Fermi-Dirac distribution, quantum oscillations encode the change in occupations of Landau levels in an energy window (of order $k_B T$) near the Fermi level. If Landau level spacings are a decreasing function of energy, then Landau levels at higher energy are more susceptible to temperature smearing. Because lower-energy Landau levels dominate the oscillation signal, the oscillation frequency (being an increasing function of energy) decreases with temperature. This explanation is already present in the main text, and we hope the referee is satisfied with it.

Incidentally, ref [27] by Koppersbusch and Fritz was never meant as an explanation for the T^2 correction in the Lifshitz-Kosevich formula. Rather, the general formula that we derive, when applied to Dirac fermions, reproduces the model-specific calculation in ref [27].

3. The quasiparticle Dirac/Weyl fermion with the linear band is a typically collective excitation mode, the hypothesized Dirac fermion with a tiny mass m_D certainly leads to a weakly-nonlinear energy dispersion, while this fact doesn't affect the topology of the band, the discussion at lines 110-116 is redundant for

the main point of the new method to detect non-trivial linear band from the trivial parabolic band.

While we agree that lines 110-116 about the tiny mass can in principle be removed from the manuscript without affecting its main message, we believe it is an important caveat to convey in the interpretation of the frequency shift. The logical possibility of the tiny mass cannot be ruled out, however unlikely the possibility. We have found in discussions with critical-minded colleagues that the conservative interpretation is well-received.

4. The authors didn't mention the top panel of figure 2 in the main text.

Thank you, this is indeed missing in the previous manuscript. We've now added the description in the caption of Fig. 2.

5. In figure 3(b), the unit of the y-axis should be shown.

Thank you for pointing this out, we realized that the definition of ρ_{osc} is not clearly addressed in the previous manuscript. Here $\rho_{\text{osc}} = \Delta\rho / \rho_{\text{BG}}$. $\Delta\rho$ is the oscillatory part of the resistivity, while ρ_{BG} stands for the polynomial fit to the magnetoresistivity as a background. Therefore ρ_{osc} is dimensionless. We've added this description to the figure caption of Fig.2.

6. The authors need to explain the different magnetic field range in three different type samples in Figure 3(b).

The magnetic field range is chosen to include as many low-noise quantum oscillations as possible, so as to improve the accuracy of frequency fitting. For Cd_3As_2 , the cyclotron mass is so low that even at 70 K, the SdH oscillation can be clearly observed in the range of 1 to 5 T. On the contrary, $\text{Bi}_2\text{O}_2\text{Se}$ has a larger cyclotron mass compared to Cd_3As_2 , therefore we took an extended field range from 2 to 10 T to get the accurate oscillation frequency at high temperatures. While for LaRhIn_5 , since the quantum oscillation frequency is as low as 6.94 T, therefore at 4 T we are already reaching the lowest Landau level and there is no point to further extend the field range as it will not include more oscillation periods. This point is addressed now in the manuscript.

7. The frequency shift is as small as 0.8% in Cd₃As₂, so the accurate fitting of frequency is significantly important in this new method, the random error of the conventional calculation method is much larger than the resolution in this manuscript. A more detailed fitting process is needed, especially about the variance and error bars.

As stated in the supplement Sec. III A, the entire experimental dataset measured at different temperatures is then fitted globally by a standard least squares regression method using the non-linear model fitting function provided by Mathematica. In this procedure, both the cyclotron mass m_c and the quantum mean free path l_q are set to be temperature-independent while the oscillation frequency F is not restricted to the same value at different temperatures, which directly yields the best-fit $\Delta F(T)$ to all experimental data. ***The error bar is determined by the standard error of the fitting parameters generated by the non-linear regression fitting procedure.***

Indeed the main text did not properly refer to the detailed discussion of this critical point! In the revised manuscript, we now give a summary of the fitting procedure in the main text, and direct the interested reader to Supplemental Sec III for more details on the fitting methodology and error bar determination.

8. The Cd₃As₂ is generalized considered a Dirac semimetal with a single Dirac cone, however, the frequency shift versus the normalized T^2 in figure 4 (a-b) showed two components in Cd₃As₂, while they showed only one in LaRhIn₅, in which a linear band and a parabolic band contribute to the quantum oscillation, please explain it.

The two components of the frequency shift are *not* related to the existence of two Fermi pockets.

Instead, the T^2 correction to the oscillation frequency is a sum of two terms, one of which is attributed to the temperature dependence of the chemical potential (at fixed carrier density), while the second term is attributed to an energy-dependent cyclotron mass. We have referred to the first term as the Sommerfeld correction, and the second as a topological correction.

For LaRhIn_5 , a multiband metal with high carrier density, the large bath of carriers from other bands effectively pinned the chemical potential to be temperature independent. The Sommerfeld contribution to $F(T)$ is therefore negligible. This means the temperature dependence of frequency is solely contributed by topological correction. Therefore it displays only one component, as shown in Fig. 4 (b).

On the other hand, for *single-frequency*, low carrier density semimetal Cd_3As_2 , there is no other band in the system to pin the chemical potential. (To clarify a potential confusion, Cd_3As_2 has *two* Dirac cones which are symmetry-related, hence each Dirac cone contributes to an oscillation with the same frequency.) Therefore the chemical potential becomes temperature-dependent and gives rise to the Sommerfeld correction to the oscillation frequency. This correction, along with the topological correction due to linear band dispersion, lead to the two components in Cd_3As_2 as displayed in Fig. 4(a).

To clearly address this point, we added emphasis on the theoretical explanation of different mechanisms for oscillation frequency shift in the revised manuscript.

9. The applicability and limitation of this new method should be summarized in the last part.

The applicability of our new method to small, topological Fermi pockets is already summarized in the Conclusion section. We have added a paragraph in the Conclusion section to elaborate on the method's limitations:

“Because the energy scale for band inversion tends to be small compared to the bandwidth, topological Fermi pockets are often small compared to the Brillouin zone. It is not impossible for highly-inverted materials that the dispersion of larger topological Fermi pockets acquires substantial nonlinear corrections, in which case our methodology becomes less useful for topological diagnosis. It is also less

straightforward to subtract the Sommerfeld correction (from the frequency shift) in materials where a topological Fermi pocket coexists with a non-topological pocket of comparable size; here, supplementing our methodology with a first-principles calculation may be useful to isolate the topological frequency shift.”

REVIEWERS' COMMENTS

Reviewer #1 (Remarks to the Author):

I have carefully read the reply and revised manuscript. In this reply, the authors have succeeded in addressing a couple of issues raised in the review processes. Unfortunately, however, the authors could not address the following point nor revise the title and statements in introduction and summary appropriately.

"Title and statements in introduction and summary is too strong and not directly related to the achievement, since the topological character needs to be inferred in the present method. They need to be thoroughly revised."

It is obvious that the topology which "must be inferred" is far from the "fingerprint". I thus cannot recommend the revised manuscript for publication in Nature Communications.

Reviewer #2 (Remarks to the Author):

This reported work demonstrates the temperature dependence of quantum oscillation frequency carries hallmark of band topology through combined experimental and theoretical studies. From quantum oscillation measurements on three representative samples, the authors reveal that the two Dirac materials with linear band dispersions exhibit universal frequency shifts consistent with the theoretical predictions, while the material with parabolic bands does not. This work establishes a new approach for distinguishing band topology via quantum oscillation measurements, which is important and can possibly facilitate the studies of topological semimetals. It should be published. All the issues raised in my first report have been addressed satisfactorily.

I have one more suggestion. I think the authors should make some discussions on whether this method is applicable if a material involves a strong Zeeman effect and/or has a small gap at Dirac nodes.

Reviewer #3 (Remarks to the Author):

I have carefully checked the authors' response letter. They almost addressed all my concerns and made proper revisions. I think that the paper is ready for the publication.

Dear Reviewers,

We thank you for taking the time to carefully review our manuscript “Temperature dependence of quantum oscillations from non-parabolic dispersions”. In the following, we will address the comments on a point-by-point basis:

REVIEWERS' COMMENTS

Reviewer #1 (Remarks to the Author):

I have carefully read the reply and revised manuscript. In this reply, the authors have succeeded in addressing a couple of issues raised in the review processes. Unfortunately, however, the authors could not address the following point nor revise the title and statements in introduction and summary appropriately. “Title and statements in introduction and summary is too strong and not directly related to the achievement, since the topological character needs to be inferred in the present method. They need to be thoroughly revised.” It is obvious that the topology which “must be inferred” is far from the “fingerprint”. I thus cannot recommend the revised manuscript for publication in Nature Communications.

Thank you for your comments. To address this point, we now have modified our title to “Temperature dependence of quantum oscillations from non-parabolic dispersions” which emphasize the detection of non-parabolic band dispersions.

Reviewer #2 (Remarks to the Author):

This reported work demonstrates the temperature dependence of quantum oscillation frequency carries hallmark of band topology through combined experimental and theoretical studies. From quantum oscillation measurements on three representative samples, the authors reveal that the two Dirac materials with linear band dispersions exhibit universal frequency shifts consistent with the

theoretical predictions, while the material with parabolic bands does not. This work establishes a new approach for distinguishing band topology via quantum oscillation measurements, which is important and can possibly facilitate the studies of topological semimetals. It should be published. All the issues raised in my first report have been addressed satisfactorily.

I have one more suggestion. I think the authors should make some discussions on whether this method is applicable if a material involves a strong Zeeman effect and/or has a small gap at Dirac nodes.

Thank you for your suggestions. In general, our methodology is applicable independent of the magnitude of the Zeeman splitting of Landau levels; this magnitude would affect the relative amplitudes of higher harmonics of quantum oscillations but not their frequency. Indeed, the analysis of the higher harmonics is the subject of a follow-up publication that we (the authors) are excited to unveil soon. It is also worth pointing out that the effective g-factor in LaRhIn_5 is large (~ 30), yet our methodology reliably identifies the linear dispersion of its small Fermi pocket. We are grateful to the referee for the specific suggestion and added a remark in the concluding paragraphs on the Zeeman effect.

Reviewer #3 (Remarks to the Author):

I have carefully checked the authors' response letter. They almost addressed all my concerns and made proper revisions. I think that the paper is ready for the publication.

Thank you again for your effort and time during the review process, which sharpen the main message of the manuscript and render the work both more rigorous and more accessible.